# The Role of Human γδ T Cells in Anti-Tumor Immunity and Their Potential for Cancer Immunotherapy

**DOI:** 10.3390/cells9051206

**Published:** 2020-05-13

**Authors:** Yuxia Liu, Cai Zhang

**Affiliations:** Institute of Immunopharmaceutical Sciences, School of Pharmaceutical Sciences, Cheeloo College of Medicine, Shandong University, 44 Wenhua West Road, Jinan 250012, China; 201816090@mail.sdu.edu.cn

**Keywords:** γδ T cells, anti-tumor effect, cancer immunotherapy

## Abstract

γδ T cells are a distinct subset of T cells whose T cell receptors consist of γ chains and δ chains, different from conventional αβ T cells. γδ T cells are considered as a member of the innate immunity because of their non-MHC restricted antigen recognition, rapid response to invading pathogens and sense early changes of malignant cells. Upon activation, they can further promote the activation of adaptive immune cells, such as T cells and B cells, by secreting various cytokines. Thus, γδ T cells are regarded as a bridge between innate immunity and acquired immunity. γδ T cells are involved in a variety of immune response processes, including immune defense and immune surveillance against infection and tumorigenesis. γδ T cells recognize multiple tumor-associated antigens or molecules in T cell receptors (TCRs)-dependent and natural killer cell receptors (NKRs)-dependent ways. γδ T cells not only display a direct killing capacity on a variety of tumors, but also exert anti-tumor immune responses indirectly by facilitating the function of other immune cells, such as dendritic cells (DCs), B cells and CD8^+^ T cells. In this review, we summarize the major subpopulations, the tumor recognition mechanisms, and the anti-tumor effects of human γδ T cells, particularly the potential of γδ T cells for cancer immunotherapy.

## 1. Introduction

Human γδ T cells are unique innate immune cells, accounting for 1–5% of lymphocytes in peripheral blood. They mainly distribute in the gut mucosa, skin and other mucosal tissues and participate in a variety of immune response and immune regulation processes, such as mediating immune inflammatory response, directly recognizing and killing tumors [1,2]. γδ T cells have gained more attention because they can quickly generate immune responses to a variety of invading pathogens and early changes of malignancy, which is likely to relate to non-MHC restricted antigen recognition, thereby, γδ T cells, together with macrophages and neutrophils, contribute to the first line of defense against foreign infections [2,3]. Upon activation, they can further promote the activation of adaptive immune cells, such as T cells and B cells, by secreting various cytokines. Thus, γδ T cells are regarded as a bridge between innate immunity and acquired immunity [4,5]. γδ T cells not only play a significant role in resisting external infections, but also play an important role in tumor immunity [2,6]. Previous studies have found that γδ T cells have powerful anti-tumor efficacy on a variety of tumors, such as breast cancer, colon cancer, lung cancer and others [7,8,9]. γδ T cells recognize tumors through T cell receptors (TCRs) and natural killer cell receptors (NKRs) [10]. On one hand, γδ T cells can directly kill tumor cells through their strong cytotoxic effects, which usually depends on their production of interferon γ (IFNγ) and tumor necrosis factor-α (TNF-α) [6]. On the other hand, they can also indirectly exert anti-tumor effects by facilitating the function of other immune cells, such as enhancing the ability of dendritic cells (DCs) to present antigens or enhancing the ability of cytotoxic T cells to kill tumor cells [11,12].

Due to the unique features of γδ T cells, such as the not MHC-restriction for tumor cell recognition and quickly production of abundant cytokines and potent cytotoxicity in response to malignancies, the anti-tumor effects of γδ T cells have demonstrated unique superiority, and γδ T cell-based cancer immunotherapy has great promise in tumor therapy [12,13]. In this review, we summarize the major characteristics of human γδ T cells, tumor cell recognition by γδ T cells, the anti-tumor mechanism of γδ T cells as well as their application and some new strategies of γδ T cells for cancer immunotherapy.

## 2. Diversity of Human γδ T Cell Subsets

Human γδ T cells can be divided into a variety of subsets based on their TCR usage, cellular phenotype and function [11,14]. (I) γδ T cell subsets classified according to the usage of TCRγ-chain or δ-chain. Generally, human γδ T cells can be divided into four major groups, Vδ1, Vδ2, Vδ3 and Vδ5 γδ T cells, based on the differences of TCR δ-chain [15,16,17] (Table 1). They have different distribution and different function. Human Vδ1^+^ γδ T cells are mainly distributed in the skin, small intestine and other mucosal tissues [18]. They are also found in small amounts in the liver and spleen [19]. Vδ1 can co-express with various Vγ chains (Vγ2, Vγ3, Vγ4, Vγ5, Vγ8 and Vγ10) to form different γδ T cell subsets [20]. Vδ1^+^ γδ T cells exhibit high anti-tumor activity against multiple cancers, such as chronic lymphoid leukemia, multiple myeloma, breast cancer, colorectal cancer and other cancers [7,18,21,22]. Vδ2^+^ γδ T cells mainly exist in peripheral blood and are the main γδ T cells involved in blood circulation. During TCR γδ rearrangement, Vδ2 is almost exclusively co-expressed with Vγ9 to form Vγ9Vδ2 T cells, which can recognize phosphoantigen and have strong anti-tumor ability against tumors such as cholangiocarcinoma, primary glioblastoma, breast cancer and other cancers [23,24,25]. Vγ9Vδ2 T cells can also inhibit tumor cell proliferation and promote tumor cell apoptosis [26]. In addition, a recent study reported that human Vδ2^+^ γδ T subpopulation includes a distinct Vγ9^-^ subset with adaptive characteristics, exerting distinct functions in microbial immunosurveillance [27]. Vδ3^+^ γδ T cells mainly exist in the liver and small intestine epithelium. There are few studies on its effect on tumor process [28]. Vδ5^+^ γδ T cells co-expressing Vγ4 recognize endothelial protein C receptor (EPCR) on cytomegalovirus (CMV)-infected epithelial cells and epithelial tumor cells via TCR [29]. (II) γδ T cell subsets classified according to γδ T cell phenotype. Human Vγ9Vδ2 T cells express some surface markers associated with functional phenotypes. They can be divided into four distinct subsets: naive, central memory, effector memory, and terminal differentiated cells, based on the cell surface expression of CD27 and CD45RA [30]. Naive (CD27^+^CD45RA^+^) and central memory (CD27^+^CD45RA^−^) Vγ9Vδ2 T cells are inclined to stay in the lymph nodes and lack immediate effector functions, while effector memory (CD27^−^CD45RA^−^) and terminally differentiated (CD27^−^CD45RA^+^) Vγ9Vδ2 T cells tend to move toward inflammatory sites and exert immediate effector functions [30]. These distinct subsets display different function in response to bacterial and viral infection. (III) γδ T cell subsets classified according to cellular function. Similar to αβ T cells, γδ T cells can be divided into different subsets based on their different functions, such as γδ T1, γδ T17, follicle-assisted γδ T (γδ Tfh), regulatory γδ T (γδ Treg) and memory γδ T cells [31]. γδ T1 cells are mainly IFN-γ-producing cells. Human peripheral blood γδ T cells (mainly Vγ9Vδ2 T cells) can produce IFN-γ in the stimulation of phosphoantigen (such as isopentenyl pyrophosphate, IPP) accompanied by IL-12 and anti-IL-4 antibodies [32]. γδ T17 cells are mainly IL-17-producing γδ T cells. IL-17 is a pro-inflammatory factor associated with the development of chronic tissue inflammation and autoimmune diseases [33]. γδ T17 cells are rarely found in healthy individuals, but accumulate in tissues with tumor and inflammation, which suggests that γδ T17 cells are associated with tumor progression [34,35]. γδ Tfh cells have similar functions to traditional Tfh cells, and can promote B cell maturation and facilitate the ability of B cells to produce antibodies [36]. γδ Treg cells are enriched in tumor infiltrating lymphocytes of several types of cancer, and can not only suppress T cell responses and DC maturation, but also induce the immunosenescence of T cells and DCs, and further amplify Treg-mediated immunosuppression [37,38]. Human Vγ9Vδ2 T cells stimulated by mycobacterium bovis bacillus Calmette–Guerin (BCG) and malaria parasite plasmodium falciparum display a memory characteristic [39,40]. Besides, Vγ9Vδ2 T cells in the presence of IL-15 and IL-21 differentiate into a memory population expressing CD45RO [41]. Whether memory γδ T cells have an effect against tumors still needs for further studies.

## 3. The Tumor Cell Recognition of γδ T Cells

### 3.1. Tumor Cell Recognition of Vγ9Vδ2 T Cells

The earliest discovery is that Vγ9Vδ2 T cells can specifically recognize non-peptide phosphoantigen, such as IPP, with a γδ TCR-dependent manner [42] (Figure 1). In normal tissues, cells synthesize isoprenoid through the mevalonate metabolism pathway to participate in the synthesis of cholesterol along with producing a small amount of IPP in the process. In order to satisfy the tumor’s energy supply in the tumor tissue, the mevalonate metabolic pathway is enhanced, leading to the accumulation of IPP exceeding the normal physiological range in the cells, and thus tumor cells are recognized and killed by Vγ9Vδ2 T cells [43]. F1-ATPase is generally expressed on the mitochondrial membrane, and occasionally can be expressed on liver cells and tumor cells. Vγ9Vδ2 T cells can recognize tumor cells expressing Fl-ATPase, which depends on the formation of complex Fl-ATPase and apolipoprotein A-I (apo A-I) [44]. Several studies have shown that Vγ9Vδ2 T cells recognize phosphoantigen depending on the butyrophilin 3A1 (BTN3A1, also known as CD277), a member of the B7 superprotein family, expressed on antigen-presenting cells (APCs) and tumor cells [45,46,47]. However, the molecular mechanism of the interaction between BTN3A1 and phosphoantigen is still unclear. More evidences support the hypothesis of “inside-out” triggering mechanism that BTN3A1 will undergo joint spatial and conformational changes when phosphoantigen binds to the B30.2 intracellular domain or juxtamembrane domain of BTN3A1 [47,48,49,50]. These spatial and conformational changes trigger the activation of Vγ9Vδ2 T cells [51]. Intriguingly, recent studies show that the other member of BTNA family, butyrophilin 2A1 (BTN2A1) is a direct ligand for Vγ9Vδ2 TCR, and is essential for phosphoantigen recognition by Vγ9Vδ2 T cells [52,53]. It is revealed that BTN2A1 and BTN3A1 are co-associated on the surface of tumor cells, and are both required for phosphoantigen recognition by Vγ9Vδ2 T cells. After phosphoantigen binds to the B30.2 intracellular domain of BTN3A1, the BTN2A1-BTN3A1 complex engages the γδ TCR via two distinct sites. BTN2A1 directly binds to the Vγ9 region of Vγ9Vδ2 TCR, while BTN3A1 binds to the Vδ2-encoded CDR2 and γ-chain-encoded CDR3 of Vγ9Vδ2 TCR [52,53]. These findings about the distinctive mode of phosphoantigen-dependent Vγ9Vδ2 T cells activation help us to better understand the recognition mechanism of γδ T cells to tumors involved with phosphoantigen and facilitate attempts to develop of γδ T cell-based cancer immunotherapies.

In addition to the TCR-dependent way, Vγ9Vδ2 T cells can recognize tumor cells through NKRs, particularly NKG2D and DNAX accessory molecule-1 (DNAM-1) (Figure 1). The NKG2D ligands currently found on human tumor cells include MHC class I chain-related molecules (MICA, MICB) and six ULBP-binding proteins (ULBP1-6) [54]. Overexpression of the ULBP1 and ULBP4 on the leukemia, lymphoma and epithelial tumors, respectively, induces cytotoxicity of Vγ9Vδ2 T cells to tumor cells [42]. In addition, DNAM-1 expressed on the Vγ9Vδ2 T cells recognizes its ligands nectin-2 and polyoma virus receptor (PVR) that are widely expressed in tumor cells, which helps Vγ9Vδ2 T cells target tumor cells [55,56].

### 3.2. Tumor Cell Recognition of Vδ1 T Cells

Like Vγ9Vδ2 T cells, Vδ1 T cells can also recognize tumor cells in TCR-dependent and NKR-dependent ways (Figure 1). Currently, very few Vδ1 TCR ligands have been described. Lipid antigens-presented by CD1d, a MHC-like molecule, are reported to bind Vδ1 TCR, revealed by the crystal structure of Vδ1 TCR in complex with CD1d-sulfatide [57,58,59]. Even though so far no clear evidence demonstrates that the involvement of CD1d-lipid antigens in tumor cell recognition by γδ T cells; however, CD1d is found to be expressed on multiple cancers, such as lymphomas, early myeloma, myeloid leukemias, medulloblastoma, and prostate cancers [60,61,62,63]. The cellular lipid composition is found to be changed during tumorigenesis, and some types of lipids, such as sulfatide, are elevated in several types of epithelial cancers [64]. Whether Vδ1 T cells may recognize specific lipid antigens presented by CD1d on the surface of tumor cells deserves further investigation. In addition, human Vδ1 T cells can recognize MICA-expressing tumor cells by direct binding of MICA to Vδ1 TCR [65]. However, the contribution of the Vδ1 TCR activation by MICA-expressing tumor cells is difficult to evaluate, because MICA also serves as a high affinity ligand for NKG2D expressed on Vδ1 T cells [13].

Vδ1 T cells can also recognize tumor cells through NKRs, which have previously been thought to be unique to NK cells [10]. NKRs include NKG2D, DNAM-1 and Natural cytotoxicity receptors (NCRs), such as NKp30, NKp44 and NKp46, and other receptors. Vδ1 T cells recognize NKG2D ligands, such as MICA/B on intestinal epithelial cells and ULBP3 expressed on B cell chronic lymphocytic leukemia (B-CLL) [66,67]. Breast-resident Vδ1 T cells are activated innately via the NKG2D receptor and are associated with the remission of triple-negative breast cancer [7]. In recent years NCR expression has been found in activated small intestinal intraepithelial lymphocytes and innate lymphoid cells (ILCs) [68]. Although Vδ1 T cells do not express NCRs under normal conditions, they can express NCRs, including NKp44, NKp30, and NKp46, under the continuous stimulation of TCR agonists in the presence of IL-15 or IL-2 [18,69]. Interestingly, this phenomenon basically appears only on Vδ1 T cells, but has not been found in Vδ2 T cells [70]. These NCR^+^ Vδ1 T cells have a stronger anti-tumor activity against tumor cells and exert high ability to secrete IFN-γ [71,72]. Although most ligands on tumors recognized by NCRs of Vδ1 T cells have not been characterized, B7-H6 has been reported as a ligand of NKp30 expressed on Vδ1 T cells [73].

### 3.3. Tumor Cell Recognition of Vδ1^−^Vδ2^−^ T Cells

Studies about Vδ1^−^Vδ2^−^ TCR ligands are rare. A previous study showed that expanded human Vδ3 T cells in vitro recognize CD1d and kill CD1d^+^ target cells [28]. A well-studied ligand of Vδ1^−^Vδ2^−^ T cells TCR is endothelial protein C receptor (EPCR), an important component of the C protein pathway, recognized by Vγ4Vδ5 TCR [29] (Figure 1). It belongs to MHC I/CD1-like protein molecules and can protect the endothelial barrier from hypoxia and inflammatory conditions [74]. The interaction of Vγ4Vδ5 TCR and EPCR may enable γδ T cells to recognize CMV-infected epithelial cells and epithelial tumor cells which up-regulation EPCR expression [29].

## 4. Anti-Tumor Mechanism of γδ T Cells

γδ T cells can exert direct anti-tumor effects through a variety of pathways. Human Vγ9Vδ2 T can kill various tumor cells depending on secretion of cytotoxic substances, such as perforin and granzyme [75,76]. Besides, human Vγ9Vδ2 T cells can kill osteosarcoma cells and chronic myeloid leukemia cells through the expression of tumor necrosis factor-related apoptosis-inducing ligand (TRAIL) [76,77] and induce apoptosis of osteosarcoma cell lines through human apoptosis-related factor ligand (FASL) [76]. Vδ1 T cells can also kill tumor cells by releasing perforin and granzyme, and exert TRAIL- and FASL-dependent cytotoxicity [78,79]. Similar to NK cells, human γδ T cells can kill tumor cells through antibody-dependent-cell-mediated cytotoxicity (ADCC) effects. CD16^+^Vγ9Vδ2 T cells can recognize monoclonal antibody coated lymphoma, chronic lymphocytic leukemia (CLL) and human epidermal growth factor receptor 2 (HER2)-positive breast cancer cells and exert anti-tumor effect via ADCC [80]. In several other studies, γδ T cells have also been shown to induce ADCC on target cells following treatment with antibodies [81,82].

γδ T cells can also exert anti-tumor effects indirectly by promoting the functions of other immune cells. A unique CXCR5^+^Vγ9Vδ2 T cell subset is able to express the costimulatory molecules ICOS and CD40L, and produces cytokines such as IL-2, IL-4, and IL-10, as well as helps B cells for the production of antibodies [83]. Vγ9Vδ2 T cells expressing CXCL13 and CXCR5 display the characteristics of follicular helper T (Tfh) cells and enhance the ability of B cells to secrete antibodies when they are co-incubated with (*E*)-4-hydroxy-3-methylbut-2-enyl pyrophosphate (HMB-PP) and IL-21 in vitro [84]. No clear evidence demonstrates that human γδ T cells exert anti-tumor function upon helping B cells for antibodies production, but several studies have shown that mouse γδ T cells can provide a protective response against tumor through providing help for B cells to class-switch to IgE production [85,86]. Besides, γδ T cells can also serve as APCs of αβ T cells and participate in the activation and proliferation of αβ T cells, so that naïve αβ T cells can differentiate into effector T cells and exert antitumor effects [87,88,89]. Furthermore, when stimulated by tumor antigens, γδ T cells will up-regulate the expression of antigen-presenting molecules HLA-DR, costimulatory and adhesion molecules (CD80, CD86, CD40) and scavenger receptor CD36, which allow γδ T cells to participate in the elimination of tumor cells [87,88]. In addition, zoledronate-activated γδ T cells can improve the cytotoxicity of NK cells, which is depended on the interaction of CD137 ligand expressed on activated γδ T cells with CD137 expressed on NK cells [90]. γδ T cells can promote the maturation of DCs. In turn, DCs can also induce the activation and proliferation of γδ T cells, assisting directly or indirectly anti-tumor immune response of γδ T cells [91,92].

## 5. γδ T Cell-Based Cancer Immunotherapy

Tumor immunotherapy restarts or enhances the anti-tumor immune response of tumor patients thereby controlling and eliminating tumors [93,94]. At present, there are a few tumor immunotherapies based on αβ T cells, but the durability and effectiveness of the treatment need to be further improved. This is because the activation of αβ T cells depends on specific tumor-associated antigens (TAA), MHC molecules, and costimulatory signals. Once TAA, MHC molecules or co-stimulatory signals lost, the tumor-killing ability of αβ T cells will be reduced and even αβ T cells will be disabled [95]. γδ T cells have no MHC restriction on tumor antigen recognition and have strong cytotoxic effects on tumor cells. Importantly, they can be easily expanded in vitro and in vivo. Therefore, γδ T cells have great application prospects in tumor immunotherapy. Currently, a number of clinical trials based on γδ T cell therapy have been finished or are ongoing to evaluate the safety and anti-tumor efficacy of γδ T cells (Table 2).

Vγ9Vδ2 T cells are the most important γδ T cells used for immunotherapy via adoptive transfer of ex vivo-expanded Vγ9Vδ2 T cells or in vivo stimulation with synthetic phosphoantigens (such as BrHPP) or aminodicarbonates (N-BPs, such as zoledronate) [96]. A series of clinical trials have completed to assess the safety and efficacy of Vγ9Vδ2 T cells for immunotherapy (Table 2). Several studies have confirmed that Vγ9Vδ2 T cells can achieve specific proliferation under the stimulation of zoledronate and IL-2 [25,97,98]. When Vγ9Vδ2 T cells are applied for cancer immunotherapy, it is often accompanied by the use of BrHPP, zoledronate and IL-2 to maintain the activation status of Vγ9Vδ2 T cells and increase the killing capacity to tumor cells [97,98,99]. Multiple clinical trials have shown that adoptive transfer of autologous Vγ9Vδ2 T cells is able to target diversified tumors, such as renal cell carcinoma, non-small-cell lung cancer, gastric cancer and others, and no severe toxicity is observed [98,100,101,102,103,104,105,106]. Six patients showed stabilized disease in ten metastatic renal cell carcinoma patients who received Vγ9Vδ2 T cells transfusion [100]. Partial or complete remissions were observed in patients who were treated with autologous Vγ9Vδ2 T cells expanded in vitro [98,104,106]. In addition, allogenic Vγ9Vδ2 T cells also have potential to be used in adoptive transfer immunotherapy. A patient with stage IV gallbladder cancer treated with allogenic Vγ9Vδ2 T cells combined with zoledronate displayed favorable results [23]. Tumor volume of the patient was reduced, importantly, the proportions of functional CD4^+^ T cells and CD8^+^ T cells increased and exhausted CD4^+^ T cells and CD8^+^ T cells decreased after treatment with Vγ9Vδ2 T cells [23]. These clinical studies suggested that adoptive transfer of autologous or allogenic Vγ9Vδ2 T cell not only exert direct γδ T cell-mediated anti-tumor effect, but also promote and rescue the anti-tumor capacity of CD4^+^ and CD8^+^ T cells, suggesting a promising immunotherapeutic strategy. Vγ9Vδ2 T cells stimulated with phosphoantigens or N-BPs in vivo have also shown potential to kill multiple tumor cells, but the clinical response rate is generally lower than adoptive transfer [25,99,107,108,109,110]. Interestingly, a recent study reported that vitamin C (VC, also known as l-ascorbic acid) and l-ascorbic acid 2-phosphate (pVC) can increase the proliferation of purified γδ T cells from zoledronate- or phosphoantigen-stimulated γδ T cells and reduce the apoptosis of γδ T cells during stimulation [111]. VC may be used to enhance the efficacy and clinical response rate of immunotherapy with Vγ9Vδ2 T cells.

Vδ1 T cells are a subpopulation with strong antitumor activity that are expected to be used in tumor immunotherapy. They can be activated by a variety of ligands, such as stress-induced autoantigens, glycolipid antigens presented by CD1d, and various potentially undiscovered ligands [112,113]. Compared with Vγ9Vδ2 T cells, Vδ1 T cells do not depend on N-BPs for their activation. Although there is only low proportion of Vδ1 T cells in peripheral blood, Vδ1 T cells can circulate and persist for a long time in the body, and have high tropism for tumor tissues, which is expected to maintain long-term therapeutic effects [112]. Vδ1 T cells can be largely expanded in vitro. Highly cytotoxic Delta one T (DOT) cells are Vδ1-enriched (>60%) γδ T cells produced by stimulation with TCR agonists and cytokines (IL-4 and IL-15) for over 3 weeks for adoptive cell therapy [69]. A Two-step method was used for DOT cells production: in the expansion stage, sorted γδ T cells were stimulated with IL-4 and TCR agonists for 2 weeks, then followed a differentiation stage with an additional week of IL-15 stimulation. This method can achieve clinical-grade 2000-fold expansion within three weeks, with the percentages of Vδ1 T increase from less than 0.5% to more than 70%. Importantly, NCRs, particularly NKp30 and NKp44, are induced, accompanied with up-regulation of NKG2D and DNAM-1 on expanded DOT cells [69]. Of note, when the in vitro expanded DOT cells were transferred in vivo, they have ability of producing abundant IFN-γ and TNF-α (but no IL-17), and displayed high capacity to inhibit tumor growth and prevent dissemination in xenograft model of CLL [69]. Vδ1 T cells have a good therapeutic effect on a variety of lymphoid and myeloid hematomas, such as CLL and acute myeloid leukemia (AML) [69,73]. The broad cytotoxicity of DOT cells against AML strongly depended on the expression of NKp30 ligand B7-H6, while DOT cells showed no reactivity against normal leukocytes [73]. Adoptive transfer of Vδ1 T cell therapy not only reduces the tumor load in the blood and target organs of patients, but also significantly extends the survival time of patients [73]. Accordingly, Vδ1 T cells, particularly DOT cells, may be a new candidate used for cancer immunotherapy depending on their strong anti-tumor efficacy and safety.

**Table 2 cells-09-01206-t002:** γδ T cell-based clinical trials.

Cell Type	Interventions	Cancers	Phase	Status	References or Study Registration
Vγ9Vδ2 T	BrHPP + IL-2 (in vitro)	RCC	I	Finished	[100]
Vγ9Vδ2 T	2M3B1PP + IL-2 (in vitro)	RCC	Pilot study	Finished	[106]
Vγ9Vδ2 T	2M3B1PP, ZOL + IL-2 (in vitro)	RCC	I	Finished	[104]
Vγ9Vδ2 T	ZOL + IL-2 (in vitro)	NSCLC	I	Finished	[101]
Vγ9Vδ2 T	ZOL + IL-2 (in vitro)	NSCLC	I	Finished	[102]
Vγ9Vδ2 T	ZOL + IL-2 (in vitro)	MM	I	Finished	[105]
Vγ9Vδ2 T	ZOL + IL-2 (in vitro)	Gastric cancer	Pilot study	Finished	[98]
Vγ9Vδ2 T	ZOL + IL-2 (in vitro)	Metastatic solid tumors	I	Finished	[103]
Vγ9Vδ2 T	Pamidronate + IL-2 (in vivo)	MM and NHL	Pilot study	Finished	[107]
Vγ9Vδ2 T	ZOL + IL-2 (in vivo)	RCC, AML and melanoma	I/II	Finished	[108]
Vγ9Vδ2 T	BrHPP + IL-2 (in vivo)	RCC, CRC and breast cancer	I	Finished	[99]
Vγ9Vδ2 T	ZOL + IL-2 (in vivo)	RCC	Pilot study	Finished	[109]
Vγ9Vδ2 T	ZOL + IL-2 (in vivo)	Breast cancer	I	Finished	[25]
Vγ9Vδ2 T	ZOL + IL-2 (in vivo)	Neuroblastoma	I	Finished	[110]
γδ T	In vitro expanded	Lymphoma and leukemia	Early phase 1	Recruiting	NCT04028440
γδ T	In vitro expanded	Acute myeloid leukemia	I	Recruiting	NCT04008381
γδ T	Combined with surgery (in vitro expanded)	Pancreatic cancer	I/II	Finished	NCT03180437
γδ T	Combined with DC-CIK cells (in vitro expanded)	Breast cancer	I/II	Finished	NCT02418481
γδ T	Combined with surgery (in vitro expanded)	Breast cancer	I/II	Finished	NCT03183206
γδ T	Combined with DC-CIK cells (in vitro expanded)	Liver cancer	I/II	Finished	NCT02425735
γδ T	Combined with surgery (in vitro expanded)	Liver cancer	I/II	Finished	NCT03183219
γδ T	Combined with DC-CIK cells (in vitro expanded)	Lung cancer	I/II	Finished	NCT02425748
γδ T	Combined with surgery (in vitro expanded)	Lung cancer	I/II	Finished	NCT03183232
γδ T	In vitro expanded	HCC	I	Not yet recruiting	NCT04032392
γδ T	Anti-CD19-CAR	Leukemia and lymphoma	I	Not yet recruiting	NCT02656147
γδ T	NKG2DL-targeting CAR	Relapsed or refractory solid tumors	I	Not yet recruiting	NCT04107142
γδ T	TEG001	AML and MM	I	Recruiting	NTR6541

RCC: renal cell carcinoma; NSCLC: non-small-cell lung cancer; MM: multiple myeloma; HCC: hepatocellular carcinoma; NHL: non-Hodgkin’s lymphoma; AML: acute myeloid leukaemia; CRC: colorectal cancer; BrHPP: bromohydrin pyrophosphate; 2M3B1PP: 2-methyl-3-butenyl-1-pyrophosphate; ZOL: zoledronate; DC-CIK cells: professional antigen- presenting dendritic cells and cytokine- induced killer cells; CAR: chimeric antigen receptor; NKG2DL: NKG2D ligand; TEG: T cells engineered with defined γδ TCRs.

## 6. New Strategies to Improve γδ T Cell-Based Cancer Immunotherapy

Currently, a number of novel strategies have been developed to improve the anti-tumor efficacy of γδ T-based immunotherapy. Bispecific antibodies have been exploited to both facilitate the activity of γδ T cells and enhance the tumor-targeting. For example, bispecific antibodies targeting HER2 and CD3 or Vγ9 chain of Vγ9Vδ2 TCR can increase γδ T cells cytotoxicity against pancreatic cancer cells [114]. Adoptive transfer of Vγ9Vδ2 T cells with the HER2/Vγ9 bispecific antibody significantly reduced the growth of pancreatic cancer and colon cancer in mouse xenograft models [114,115]. In addition, bispecific antibody targeting both Vγ9Vδ2 TCR and epidermal growth-factor receptor (EGFR) allows Vγ9Vδ2 T cells accumulation and activation specifically at the tumor sites, and kills tumor cells in vitro and in mouse xenograft model [116]. Furthermore, a (HER2)2xCD16 tribody form, which comprises two HER2-specific single chain fragment variable fused to a fragment antigen binding to the CD16 expressed on γδ T or NK cells, enhanced the anti-tumor activity of γδ T and NK cells against HER2-expressing tumors, including pancreatic cancer, breast cancer and ovarian cancer [117]. Thus, bispecific antibodies may represent an attractive strategy to enhance anti-tumor cytotoxicity of γδ T cells.

In recent years, CAR-T cells have made great progress in the treatment of acute leukemia and non-Hodgkin’s lymphoma. Therefore, CAR-T cells are considered to be one of the most promising tumor immunotherapies [118]. However, CAR-T cell immunotherapy still has some problems in the clinical application, such as cytokine storm, off-target effect, poor effect on solid tumors and a high recurrence rate after treatment [119]. Due to the unique ability of γδ T cells to recognize tumors, people have begun to consider whether CAR-γδ T cells can become another effector cells with great potential for tumor adoptive cell immunotherapy [120]. Previous study has confirmed that targeting CD19 of CAR-γδ T cells have significantly improved the cytotoxicity of CD19^+^ tumor cells in vitro and in vivo, indicating that CAR-γδ T cells are expected to be used to tumor immunotherapy [121]. GD2 is a ganglioside that is widely expressed on the surface of neuroblastoma cells and several other cancer types. Targeting GD2 CAR-γδ T cells are designed in order to limit the toxic and side effects of CAR-γδ T cells [122]. In this design, the activation of CAR-γδ T cells requires two independent signals. γδ TCR recognizes the tumor antigen as the first signal. Then the monoclonal antibody against GD2 recognizes GD2 and activates the downstream co-stimulation signal domain to provide a second signal. CAR-γδ T cells can be activated to exert antitumor effects with the simultaneous participation of these two signals [122]. Since normal cells will not be recognized by γδ T cells, modified GD2-targeting CAR-γδ T cells do not have cytotoxicity to normal tissues. Other study also showed that their constructed GD2-targeted CAR-γδ T cells can maintain the ability to migrate to tumor sites and tumor cell-specific cytotoxicity [123]. CD19-targeting CAR-γδ T cells are being developed for the treatment of lymphoma and leukemia in an ongoing clinical trial. A clinical trial of NKG2D ligand-targeting CAR-γδ T cells is being implemented for the treatment of relapsed or refractory solid tumors (Table 2). CAR-γδ T cells are expected to become a new type of tumor immunotherapy in the future.

A new type of CAR-T cells, named T cells engineered with defined γδ TCRs (TEGs), has recently produced by transducing high-affinity Vγ9Vδ2 TCR into αβ T cells [124,125]. This strategy combines the advantages of γδ T cells that target tumors through Vγ9Vδ2 TCR and retains ability of high proliferation, with the memory features of conventional αβ T cells. Therefore, TEGs can target a large broad of tumor cells based on the broad reactivity of Vγ9Vδ2 TCR, and can overcome the low persistence and impaired activation of γδ T cells in tumor microenvironment [12,13]. It is reported that TEG exhibits highly cytotoxicity against both haematological and solid tumors, including leukemia stem cells [125,126]. Currently, TEGs product has been produced under GMP conditions and are being used in a Phase I clinical trials for treatment of patients with relapsed and refractory AML and multiple myeloma (NTR6541) [127].

Checkpoint inhibitors, particularly anti-cytotoxic T lymphocyte antigen 4 (CTLA-4) and anti- programmed cell death 1 (PD-1)/programmed cell death-ligand (PD-L1) antibodies, have shown promising clinical efficacy in treatment of multiple tumors [128,129,130,131]. Several studies have found that Vγ9Vδ2 T cells express some inhibitory receptors (such as PD-1 and TIM-3) [132,133]. The presence of TIM-3 suppress the killing efficacy of Vγ9Vδ2 T cells by inhibiting the expression of perforin and granzyme B [132]. Blocking PD-1 and TIM-3 can rescue the anti-tumor activity of Vγ9Vδ2 T cells. Moreover, PD-1 blockade boosts γδ T cell-mediated ADCC in follicular lymphoma [133]. Combination therapy with adoptive transfer of γδ T cells and checkpoint receptor blockade could be a new strategy for γδ T cell-based cancer immunotherapy.

## 7. Conclusions and Perspectives

The anti-tumor effects of γδ T cells and their great potentiality for tumor immunotherapy have gradually attracted more and more attention. Because γδ T cells have no MHC restriction for tumor antigen recognition, γδ T cells-based tumor immunotherapy may be able to avoid immune escape caused by down-regulation of MHC class I molecules and also avoid the graft-versus-host effects (GVHD) of MHC-mismatched αβ T cells [12,134]. It is suggested that γδ T cells might be particularly suited for the therapy of tumors with low mutational burdens, where immune checkpoint inhibition is usually unsuccessful. γδ T cells have the potential to become emerging allogeneic cell therapy products in the future. However, there are still some problems in using γδ T cells for the clinical application. One of the problems is the low response rate of treatment of Vγ9Vδ2 T in clinal trails. Engineered γδ T cells (such as CAR-γδ T cells and TEGs) or in combination with checkpoint inhibitors may improve the response rate of γδ T cell therapy and have a better anti-tumor efficacy. New methods or techniques need to be established to achieve effective expansion of Vγ9Vδ2 T cells and Vδ1 T cells. Although Vγ9Vδ2 T cells activated by zoledronate and low-dose IL-2 are confirmed to be safe and well tolerated, the clinical outcomes are limited [25]. Other cytokines, such as IL-15, may be superior to IL-2 for the activation and persistence of Vγ9Vδ2 T cells in vivo [135,136]. Currently, the major of clinical trials only focus on autologous γδ T cells, development of allogenic off-the-shelf γδ T cells are needed and feasible because the unique features of γδ T cells, such as non-MHC restriction and no risk of GVHD. Besides, the mechanism of tumor-associated antigens or ligands recognized by γδ T cells via their TCRs or NKRs needs to be further understood, which may help us find some new targets for tumor immunotherapy.

## Figures and Tables

**Figure 1 cells-09-01206-f001:**
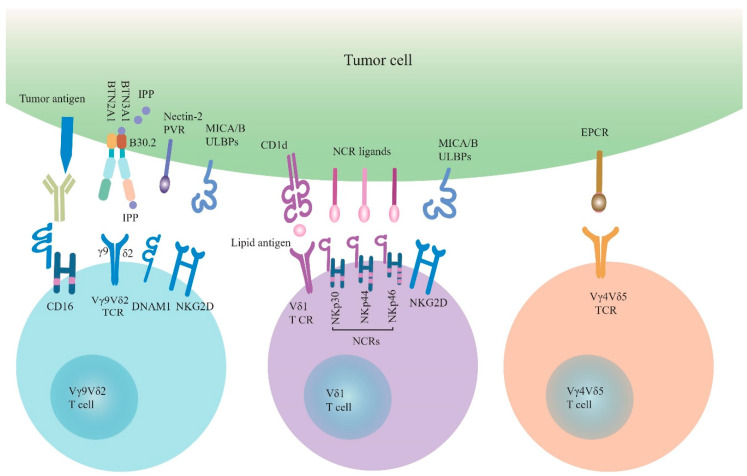
The tumor cell recognition of γδ T cells. γδ T cells recognize tumor cells through γδ TCRs and NKRs. Vγ9Vδ2 TCR recognizes pAg (such as IPP) depending on BTN3A1 and BTN2A1. Besides, Vγ9Vδ2 T cells also recognize tumor cells through NKG2D and DNAM-1, which engage with their ligands (MICA/B, ULBPs and Nectin-2, PVR). Vγ9Vδ2 T cells express CD16 that binds therapeutic antibodies to engage Vγ9Vδ2-mediated ADCC. Vδ1 T cells recognize lipid antigen-presented by CD1d through Vδ1 TCR. NKG2D and NCRs (NKp30, NKp44, NKp46) with their ligands are involved in the tumor cell recognition by Vδ1 T cells. In addition, EPCR as a ligand of Vγ4Vδ5 TCR is involved in the tumor cell recognition by Vγ4Vδ5 T cells. TCRs: T cell receptors; NKRs: natural killer cell receptors; pAg: phosphoantigen; IPP: isopentenyl pyrophosphate; BTN3A1: butyrophilin 3A1; BTN2A1: butyrophilin 2A1; DNAM-1: DNAX accessory molecule 1; PVR: polyoma virus receptor; MICA/B: MHC class I-related chain A/B; ULBPs: UL16-binding proteins; ADCC: antibody-dependent cell-mediated cytotoxicity; NCRs: natural cytotoxicity receptors; EPCR: endothelial protein C receptor.

**Table 1 cells-09-01206-t001:** Features of human γδ T cell subsets.

Subset	Common Vγ Pairs	Tissue Distribution	Anti-Tumor Effector Molecules
Vδ1	Vγ2, Vγ3, Vγ4, Vγ5, Vγ8, Vγ10	Skin, small intestine, liver, spleen	IFN-γ, TNF-α, Perforin, Granzyme, TRAIL, FASL
Vδ2	Vγ9	Peripheral blood	IFN-γ, TNF-α, Perforin, Granzyme, TRAIL, FASL
Vδ3	Not defined	Liver, small intestine	Not defined
Vδ5	Vγ4	Peripheral blood	Not defined

TRAIL: tumor necrosis factor-related apoptosis-inducing ligand; FASL: human apoptosis-related factor ligand.

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
