# Peer review of "The Role of Human γδ T Cells in Anti-Tumor Immunity and Their Potential for Cancer Immunotherapy"

_cells, 2020, doi:10.3390/cells9051206_

Round 1
Reviewer 1 Report
Liu and Zhang review the potential of gd T cells for anti-tumor therapy. The manuscript is ok, but could be sharpened by addressing the following minor issues:
Abstract:
- For readers outside of the field, the expression “As a bridge between innate and adaptive immunity …” should be explained.
- Also, the statement “ … the presence of gd T cells in the tumor … linked to poor prognosis …” may be easily misunderstood. Please explain that different subsets of gd T cells in the tumor have different impact on outcome. (Same issue line 36.)
Introduction:
- line 29 why first line of defense? Macrophages and neutrophils should be even more first liners?
- There are only 5 citations in the introduction
- The authors describe CD27, Ly6C, CD44 as surface markers for different gd T cell subsets without giving main citations.
- Diversity of gd T cell subsets:
- A table (classification, phenotype and function in tumor immunology, etc. ) would be useful to guide the reader through all the following review sections.
- line 64 “There are many …” this sentence could be more specific.
- The FOXP3/ gd Treg part is controversial to my knowledge
- line 76 only one seemingly random mouse gd T cell reference to cover gd T cell memory is not sufficient to cover this topic.
- Anti-tumor efficacy of gd T cells
- line 94 – 99: Here, BTN2A1 as a second major ligand of Vg9Vd2 T cells should be mentioned (doi: 10.1126/science.aay5516 and doi: 10.1016/j.immuni.2020.02.014.)
- line 80 – 123: It would be helpful to correlate mechanisms of tumor antigen recognition to the gd T cell subsets defined in section 2.
- reference 45 is a retraction and should not be cited here
- line 124 – 171. please clarify whether mouse or human gd T cell subsets are discussed.
- gd T cell-based tumor immunotherapy (check subheading)
- This section is the strongest part of the manuscript. However, the work of Kuball and colleagues on TEG (T cells engineered to express gd TCR) is not covered. See recent review in NRDD https://doi.org/10.1038/s41573-019-0038-z
- line 263 Why is this particular name mentioned here? (just style?)
In Figure 1, why is the tumor cell on top bigger than at the on at the bottom?. Consider revising.
Author Response
Reviewer 1:
Liu and Zhang review the potential of γδ T cells for anti-tumor therapy. The manuscript is ok, but could be sharpened by addressing the following minor issues:
Abstract:
- For readers outside of the field, the expression “As a bridge between innate and adaptive immunity …” should be explained.
Response: Thanks for the comments of reviewer. We explain the reasons of “As a bridge between innate and adaptive immunity” as followed: γδ T cells are considered as a member of the innate immunity because of their non-MHC restricted antigen recognition and rapid response to invading pathogens. Upon activation, they can further promote the activation of adaptive immune cells, such as T cells and B cells, by secreting various cytokines. Thus, γδ T cells are regarded as a bridge between innate immunity and acquired immunity. We revised this section in Abstract。
- Also, the statement “ … the presence of γδ T cells in the tumor … linked to poor prognosis …” may be easily misunderstood. Please explain that different subsets of γδ T cells in the tumor have different impact on outcome. (Same issue line 36.)
Response: Thanks. According to the suggestion from reviewer 2, we deleted these sentences.
Introduction:
- line 29 why first line of defense? Macrophages and neutrophils should be even more first liners?
Response: Thanks for your comments. We explained as followed. γδ T cells have gained more attention because they can quickly generate immune responses to a variety of invading pathogens and early changes of malignancy, which is likely to relate to non-MHC restricted antigen recognition, thereby, γδ T cells, together with macrophages and neutrophils, contribute to the first line of defense against foreign infections
- There are only 5 citations in the introduction
Response: Thanks. We supplemented some important references in this section. It includes 13 citations in introduction in this version.
- The authors describe CD27, Ly6C, CD44 as surface markers for different γδ T cell subsets without giving main citations.
Response: Thanks. According to the suggestion from reviewer 2, we deleted this sentence because CD27 and Ly-6C are markers of mouse γδ T cells.
Diversity of γδ T cell subsets:
- A table (classification, phenotype and function in tumor immunology, etc. ) would be useful to guide the reader through all the following review sections.
Response: Thanks. We supplemented Table 1 to summarize the classification, phenotype and function.
- line 64 “There are many …” this sentence could be more specific.
Response: Thanks. We deleted the description of murine γδ T cell subsets as suggested by reviewer 2. We describe the four human γδ T cell subsets based on phenotypes of CD27 and CD45RA in detailed as followed:
Human Vγ9Vδ2 T cells express some surface marker associated with functional phonotypes. They can be divided into four distinct subsets: naive, central memory, effector memory, and terminal differentiated cells, based on the cell surface expression of CD27 and CD45RA. Naive (CD27+CD45RA+) and central memory (CD27+CD45RA-) Vγ9Vδ2 T cells are inclined to stay in the lymph nodes and lack of immediate effector functions, while effector memory (CD27-CD45RA-) and terminally differentiated (CD27-CD45RA+) Vγ9Vδ2 T cells tend to move toward inflammatory sites and exert immediate effector functions. These distinct subsets display different function in response to bacterial and viral infection.
- The FOXP3/ γδ Treg part is controversial to my knowledge
Response: We revised the description of gδ Treg cells as followed: gδ Treg cells are enriched in tumor infiltrating lymphocytes of several types of cancer, and can not only suppress T cell responses and DC maturation, but also induce the immunosenescence of T cells and DCs, and further amplify Treg-mediated immunosuppression.
- line 76 only one seemingly random mouse γδ T cell reference to cover γδ T cell memory is not sufficient to cover this topic.
Response: We deleted the description about mouse gδ T cells as suggested by reviewer 2.
Anti-tumor efficacy of γδ T cells
- line 94 – 99: Here, BTN2A1 as a second major ligand of Vg9Vd2 T cells should be mentioned (doi: 10.1126/science.aay5516 and doi: 10.1016/j.immuni.2020.02.014.)
Response: Thanks a lot. We supplemented the description about the BTN2A1 as a second major ligand of Vγ9Vδ2 T cells, as followed:
Intriguingly, recent studies show that the other member of BTNA family, butyrophilin 2A1 (BTN2A1) is a direct ligand for Vγ9Vδ2 TCR, and is essential for phosphoantigen recognition by Vγ9Vδ2 T cells. It is revealed that BTN2A1 and BTN3A1 are co-associated on the surface of tumor cells, and are both required for phosphoantigen recognition by Vγ9Vδ2 T cells. After phosphoantigen binds to the B30.2 intracellular domain of BTN3A1, the BTN2A1-BTN3A1 complex engages the γδ TCR via two distinct sites. BTN2A1 directly binds to the Vγ9 region of Vγ9Vδ2 TCR, while BTN3A1 binds to the Vd2-encoded CDR2 and g-chain-encoded CDR3 of Vγ9Vδ2 TCR.
- line 80 – 123: It would be helpful to correlate mechanisms of tumor antigen recognition to the γδ T cell subsets defined in section 2.
Response: Thanks. We have described the correlate mechanisms of tumor cell recognition upon the γδ T cell subsets defined in section 2.
- reference 45 is a retraction and should not be cited here
Response: Thanks. We are very sorry for our mistake and carelessness. We corrected.
- line 124 – 171. please clarify whether mouse or human γδ T cell subsets are discussed.
Response: We deleted the description about mouse gδ T cells as suggested by reviewer 2.
γδ T cell-based tumor immunotherapy (check subheading)
- This section is the strongest part of the manuscript. However, the work of Kuball and colleagues on TEG (T cells engineered to express γδ TCR) is not covered. See recent review in NRDD https://doi.org/10.1038/s41573-019-c0038-z
Response: Thanks a lot. We supplemented the description about TEG in a new section: “New strategies to improve γδ T cell-based cancer immunotherapy”. It is described as followed: A new type of CAR-T cells, named T cells engineered with defined γδ TCRs (TEGs), has recently produced by transducing high-affinity Vγ9Vδ2 TCR into αβ T cells. This strategy combines the advantages of γδ T cells that target tumors through Vγ9Vδ2 TCR and retain ability of high proliferation, with the memory features of conventional αβ T cells. Therefore, TEGs can target a large broad of tumor cells based on the broad reactivity of Vγ9Vδ2 TCR, and can overcome the low persistence and impaired activation of γδ T cells in tumor microenvironment. It is reported that TEG exhibits highly cytotoxicity against both haematological and solid tumors, including leukemia stem cells. Currently, TEGs product has been produced under GMP conditions and are being used in a Phase I clinical trials for treatment of patients with relapsed and refractory AML and multiple myeloma (NTR6541).
- line 263 Why is this particular name mentioned here? (just style?)
Response: We have changed the description of the sentence as followed. A stage IV gallbladder cancer treated with allogenic Vγ9Vδ2 T cells combined with zoledronate displayed favorable results. Tumor volume of patients was reduced, importantly, the proportions of functional CD4+ T cells and CD8+ T cells increased and exhausted CD4+ T cells and CD8+ T cells decreased after treatment with Vγ9Vδ2 T cells.
In Figure 1, why is the tumor cell on top bigger than at the on at the bottom?. Consider revising.
Response: Thanks. We drew a new figure describing the tumor cell recognition of γδ T cells.
Reviewer 2 Report
Overall, the review article has many flaws, which outweigh the positives aspects of the text. It reads like a list of studies without any perspective or insight from the authors. The explanation of the studies included is superficial. Many concepts and acronyms are not explained, making the manuscript inaccessible to a non-expert. Several publications are misquoted. New publications – those published within the past two years – were not included. The article lacks balance, specifically the limitations of gamma delta T cell immunotherapy and their killing ability. The figures are very similar to other published articles.
To improve the quality of this manuscript, the authors should help the reader digest the information by providing commentary and personal opinion. It is clear that their interest (and bias) is focused on the anti-tumor role of gamma delta T cells, so I recommend that the pro-tumor section be omitted. In this way, the authors could expand on the anti-tumor properties of the cells and include more information on model systems. And the focus should be on human cells, leaving out mouse studies.
Below are more specific suggestions and comments written by section:
- Introduction:
- Line 29/30: right, but jump too soon to cancer (causality within the sentence?)
- Line 41: define MDSC and explain what these groupof cells refer to. It should be ‘myeloid-derived suppressor cells’ not ‘bone marrow-derived suppressor cells’
- Line 41: more citations to prove recruitment of MDSCs (the reference does not explicitly prove their statement)
- Line 44: wording here is incorrect. Tumor microenvironment does not convert on subpopulation into another
- Diversity of subsets
- The numbers in parentheses throughout the paragraph is not proper English and confusing to the reader
- Line 51: sentence doesn’t sound complete (structuring of paragraph -> poor, not clear)
- Line 53: reference is misquoted. Maybe Raverdeau et al. 2019 better reference
- Line 55: reference doesn’t show what is stated
- Line 58: wording (almost what? exclusively?)
- Line 65: human, mouse or both markers?
- Line 65: Ly-6C? This needs a reference. I do not know of any study showing that Ly6C is a marker for gdT cells
- Line 66: mouse studies are out of place here. Needs context
- Line 69/70: misquoted Thone 2014 paper – no mention of gdT cells in this paper. CD27 is not a marker of memory cells. Do the authors mean CCR7?
- Need to quote papers by Silva-Santos lab here when discussing CD27 (Ribot 2009)
- Line 73: language “aremainly secrete..”
- Anti-tumor efficacy of gdT cells
- Line 83: isopentenyl typo (-ene ending for unsaturated hydrocarbons, -yl ending for hydroxyl group)
- Line 90-93: Sentence too long and unnaturally structured (order of some parts seems grammatically incorrect), does reference show what they say? In paper zoledronate is used in high doses to render cells apoptotic and make them upregulate F1-ATPase on tumour cells, also no mention of complex partner apoA-1 in this reference.
- Discuss recent Ben Willcox paper by M.M. Karunakaran, March 2020, “Butyrophilin-2A1 directly binds germline-encoded regions of the Vg9Vd2 TCR and is essential for phosphoantigen sensing”
- Line 102: randomly jump to next example. Maybe a transition like “Another well-studied mechanism…”
- Line 110: plural for NKG2D ligands, more than one out there
- Line 114: change from NKR to NCR is confusing for reader
- Line 117: no reference to back up statement
- Anti-tumor mechanism
- Line 132: the references don’t mention Granzyme B – misquoted
- Line 133: not the key message of that reference, only shown in fig 1c and not further described in text of paper.
- Line 165: it’s HLA-DR not -DA
- Line 167: misquoted reference
- Line 168: this is not the primary reference for their statement (original: Maniar A, Zhang X, Lin W, et al. Human gammadelta T lymphocytes induce robust NK cell-mediated antitumor cytotoxicity through CD137 engagement., Blood, 2010, vol. 116)
Figure 1
- be specific about which sub-population this refers to and which species. Not all sub-populations will express the same molecules. Do you mean MHC-II instead of MHC-I? GM-CSF is not listed in cartoon but referenced in figure legend. How do gdT cells promote class switching?
- Role of gdT in tumor promotion
- The authors should omit this section
- 176: Does the TME actively transform gdT cells? Subpopulations exist without influences of TME, so TME probably just favours proliferation/expansion/survival of distinct subsets, but more passively.
- 181: list needs more (primary) references? The authors miss several papers from Silva-Santos, de Visser, Sears labs
- Figure 2: IL-17 not just effect on angiogenesis -> neutrophil recruitment, gdTreg nomenclature is not well established
- 213: miR-99-5p confusing, is it actually the other one? “miR-99a-5p had no effect on the expression of CD107a, granzyme B and perforin in γδ T cells. Taken together, these results suggest that miR-125b-5p regulates the cytotoxicity of γδ T cells by targeting degranulation and the expression of IFN-γ and TNF-α.” But quite confusing in reference itself as well…so not sure.
- 218: Daley/Miller paper as reference for human patient statement, but paper mainly mouse -> maybe better reference out there? (same for rest of paragraph FoxP3/PD-L1)
Figure 2
- This figure should be omitted
- It is very similar to one published in Nature Reviews Cancer by Silva-Santos
- immunotherapy
- 243-245: brief explanation of which specific gdT cell subpopulation/modification strategy (CAR, transduction with NKR, DOT…) has been exploited for individual trial and outcome of the trial?
- 251: also add a reference?
- 254: “Several studies..” but then just refer to one, following sentence just a repetition?
- 268: reference 79 is not about Vg9Vd2 -- misquoted
- 273: frequencies of…(2x)
- 277: wording (concerned?)
- 284: name DOT protocol, established method?
- Conclusion
- repetitive
- Leave out microbiota stuff
Author Response
Overall, the review article has many flaws, which outweigh the positives aspects of the text. It reads like a list of studies without any perspective or insight from the authors. The explanation of the studies included is superficial. Many concepts and acronyms are not explained, making the manuscript inaccessible to a non-expert. Several publications are misquoted. New publications – those published within the past two years – were not included. The article lacks balance, specifically the limitations of gamma delta T cell immunotherapy and their killing ability. The figures are very similar to other published articles.
To improve the quality of this manuscript, the authors should help the reader digest the information by providing commentary and personal opinion. It is clear that their interest (and bias) is focused on the anti-tumor role of gamma delta T cells, so I recommend that the pro-tumor section be omitted. In this way, the authors could expand on the anti-tumor properties of the cells and include more information on model systems. And the focus should be on human cells, leaving out mouse studies.
Response: We are really appreciate your hard working and valuable suggestions. As suggested, we expanded on the anti-tumor properties of the γδ T cells, γδ T-based cancer immunotherapy and multiple new strategies. We deleted the sections of pro-tumor effect of γδ T cells and data about mouse γδ T cells. In addition, we supplemented some commentaries and personal opinions for helping readers understand. We revised the manuscript and corrected the mistakes, including some inaccurate descriptions and some misquoted publications, carefully.
Below are more specific suggestions and comments written by section:
Introduction:
- Line 29/30: right, but jump too soon to cancer (causality within the sentence?)
Response: Thanks. We have added some descriptions in front of this sentence and explained the causality within the sentence as followed: γδ T cells have gained more attention because they can quickly generate immune responses to a variety of invading pathogens and early changes of malignancy, which is likely to relate to non-MHC restricted antigen recognition, thereby, γδ T cells, together with macrophages and neutrophils, contribute to the first line of defense against foreign infections. Upon activation, they can further promote the activation of adaptive immune cells, such as T cells and B cells, by secreting various cytokines. Thus, γδ T cells are regarded as a bridge between innate immunity and acquired immunity.
- Line 41: define MDSC and explain what these group of cells refer to. It should be ‘myeloid-derived suppressor cells’ not ‘bone marrow-derived suppressor cells’
Response: Thanks. We deleted this sentence as suggested because it is about pro-tumor function of γδ T cells.
- Line 41: more citations to prove recruitment of MDSCs (the reference does not explicitly prove their statement)
Response: Thanks. We deleted this sentence as suggested because it is about pro-tumor function of γδ T cells.
- Line 44: wording here is incorrect. Tumor microenvironment does not convert on subpopulation into another
Response: Thanks. We deleted this sentence as suggested because it is about pro-tumor function of γδ T cells.
Diversity of subsets
- The numbers in parentheses throughout the paragraph is not proper English and confusing to the reader
Response: Thanks a lot. The numbers have been changed as suggested.
- Line 51: sentence doesn’t sound complete (structuring of paragraph -> poor, not clear)
Response: Thanks. We revised the sentence as followed: Human γδ T cells can be divided into a variety of subsets based on their TCR usage, cellular phenotype and function. (I) γδ T cell subsets classified according to the usage of TCRγ-chain or δ-chain. Similar descriptions about other γδ T cell subsets classification were also revised.
- Line 53: reference is misquoted. Maybe Raverdeau et al. 2019 better reference
Response: Thanks. We changed the description of this sentence and quoted another appropriate reference (Sullivan L.C, Shaw E.M, Stankovic S, et al. The complex existence of γδ T cells following transplantation: the good, the bad and the simply confusing. Clinical & translational immunology 2019, 8, e1078.).
It is described as followed: γδ T cell subsets classified according to the usage of TCRγ-chain or δ-chain. Generally, human γδ T cells can be divided into four major groups, Vδ1, Vδ2, Vδ3 and Vδ5 γδ T cells, based on the differences of TCR δ-chain.
- Line 55: reference doesn’t show what is stated
Response: Thanks. We have changed the description of this sentence and changed the reference (Sant S, Jenkins M.R, Dash P, et al. Human γδ T-cell receptor repertoire is shaped by influenza viruses, age and tissue compartmentalisation. Clinical & translational immunology 2019, 8, e1079.).
It is described as followed: Vδ1 can co-express with various Vγ chains (Vγ2, Vγ3, Vγ4, Vγ5, Vγ8 and Vγ10) to form different γδ T cell subsets.
- Line 58: wording (almost what? exclusively?)
Response: Thanks. This has been changed as followed: During TCRγδ rearrangement, Vδ2 is almost exclusively co-expressed with Vγ9 to form Vγ9Vδ2 T cells.
- Line 65: human, mouse or both markers?
Response: Thanks. We deleted this sentence because CD44 and Ly-6C are markers of mouse γδ T cells.
- Line 65: Ly-6C? This needs a reference. I do not know of any study showing that Ly6C is a marker for γδ T cells
Response: Thanks. This sentence was deleted because Ly-6C is a marker of mouse γδ T cells. Reference: Lombes A, Durand A, Charvet C, et al. Adaptive Immune-like γ/δ T Lymphocytes Share Many Common Features with Their α/β T Cell Counterparts. Journal of immunology (Baltimore, Md: 1950) 2015, 195, 1449-1458.
- Line 66: mouse studies are out of place here. Needs context
Response: We deleted mouse studies in here as suggested.
- Line 69/70: misquoted Thone 2014 paper – no mention of γδ T cells in this paper. CD27 is not a marker of memory cells. Do the authors mean CCR7?
Response: We are very sorry for our careless. This reference was corrected as the following: Qin, G, Liu, Y, Zheng, J, et al. Phenotypic and functional characterization of human γδ T-cell subsets in response to influenza A viruses. The Journal of infectious diseases 2012, 205, 1646-1653.
- Need to quote papers by Silva-Santos lab here when discussing CD27 (Ribot 2009)
Response: We did not quote this paper because it is a study about mouse γδ T cells. We quoted another appropriate paper as followed.
Qin, G, Liu, Y, Zheng, J, et al. Phenotypic and functional characterization of human γδ T-cell subsets in response to influenza A viruses. The Journal of infectious diseases 2012, 205, 1646-1653.
- Line 73: language “aremainly secrete..”
Response: We changed the description of this sentence as followed. γδ T1 cells are mainly IFN-γ-producing cells.
Anti-tumor efficacy of γδ T cells
- Line 83: isopentenyl typo (-ene ending for unsaturated hydrocarbons, -yl ending for hydroxyl group)
Response: We are very sorry for our mistake. We have corrected the mistake.
- Line 90-93: Sentence too long and unnaturally structured (order of some parts seems grammatically incorrect), does reference show what they say? In paper zoledronate is used in high doses to render cells apoptotic and make them upregulate F1-ATPase on tumour cells, also no mention of complex partner apoA-1 in this reference.
Response: We changed the reference (Scotet E, Martinez L.O, Grant E, et al. Tumor recognition following Vgamma9Vdelta2 T cell receptor interactions with a surface F1-ATPase-related structure and apolipoprotein A-I. Immunity 2005, 22, 71-80.). We redescribed this sentence as followed: Vγ9Vδ2 T cells can recognize tumor cells expressing Fl-ATPase, which depends on the formation of complex Fl-ATPase and apolipoprotein A-I (apo A-I).
- Discuss recent Ben Willcox paper by M.M. Karunakaran, March 2020, “Butyrophilin-2A1 directly binds germline-encoded regions of the Vg9Vd2 TCR and is essential for phosphoantigen sensing”
Response: Thanks a lot. We supplemented discussion about butyrophilin-2A1 as followed. Intriguingly, recent studies show that the other member of BTNA family, butyrophilin 2A1 (BTN2A1) is a direct ligand for Vγ9Vδ2 TCR, and is essential for phosphoantigen recognition by Vγ9Vδ2 T cells. It is revealed that BTN2A1 and BTN3A1 are co-associated on the surface of tumor cells, and are both required for phosphoantigen recognition by Vγ9Vδ2 T cells. After phosphoantigen binds to the B30.2 intracellular domain of BTN3A1, the BTN2A1-BTN3A1 complex engages the γδ TCR via two distinct sites. BTN2A1 directly binds to the Vγ9 region of Vγ9Vδ2 TCR, while BTN3A1 binds to the Vd2-encoded CDR2 and g-chain-encoded CDR3 of Vγ9Vδ2 TCR.
- Line 102: randomly jump to next example. Maybe a transition like “Another well-studied mechanism…”
Response: Thanks. We changed description this sentence as followed. A well-studied ligand of Vδ1- Vδ2- T cells TCR is endothelial protein C receptor (EPCR), an important component of the C protein pathway, recognized by Vγ4Vδ5 TCR.
- Line 110: plural for NKG2D ligands, more than one out there
Response: Thanks. We corrected.
- Line 114: change from NKR to NCR is confusing for reader
Response: Thanks. In manuscript, we explained NCR, which is acronym of natural cytotoxicity receptor.
- Line 117: no reference to back up statement
Response: Thanks. We supplemented references:
Mikulak J, Oriolo F, Bruni E, et al. NKp46-expressing human gut-resident intraepithelial Vδ1 T cell subpopulation exhibits high antitumor activity against colorectal cancer. JCI insight 2019, 4.
Almeida A.R, Correia D.V, Fernandes-Platzgummer, A, et al. Delta One T Cells for Immunotherapy of Chronic Lymphocytic Leukemia: Clinical-Grade Expansion/Differentiation and Preclinical Proof of Concept. Clinical cancer research: an official journal of the American Association for Cancer Research 2016, 22, 5795-5804.
Anti-tumor mechanism
- Line 132: the references don’t mention Granzyme B – misquoted
Response: Thanks. We changed the description as followed: Human Vγ9Vδ2 T can kill various tumor cells depending on secretion of cytotoxic substances, such as perforin and granzyme.
We quoted references as followed:
Stam A, de Bruin R, Roovers R et al. Specific tumor targeting and activation of Vγ9Vδ2 T cells by bi-specific nanobodies. Journal for ImmunoTherapy of Cancer 2014, 2.
Li Z, Peng H, Xu Q et al. Sensitization of human osteosarcoma cells to Vγ9Vδ2 T-cell-mediated cytotoxicity by zoledronate. Journal of orthopaedic research: official publication of the Orthopaedic Research Society 2012, 30, 824-830.
- Line 133: not the key message of that reference, only shown in fig 1c and not further described in text of paper.
Response: We deleted this sentence because it is involved in mouse γδ T cells.
- Line 165: it’s HLA-DR not -DA
Response: We are very sorry for our careless. We have corrected the spelling mistake.
- Line 167: misquoted reference
Response: We are very sorry for our mistake. We corrected the reference to the following: Maniar A, Zhang X, Lin W, et al. Human gammadelta T lymphocytes induce robust NK cell-mediated antitumor cytotoxicity through CD137 engagement., Blood, 2010, vol. 116.
- Line 168: this is not the primary reference for their statement (original: Maniar A, Zhang X, Lin W, et al. Human gammadelta T lymphocytes induce robust NK cell-mediated antitumor cytotoxicity through CD137 engagement., Blood, 2010, vol. 116)
Response: Thanks a lot. We have quoted the reference as suggested.
Figure 1
be specific about which sub-population this refers to and which species. Not all sub-populations will express the same molecules. Do you mean MHC-II instead of MHC-I? GM-CSF is not listed in cartoon but referenced in figure legend. How do γδ T cells promote class switching?
Response: Thanks. We drew a new figure describing the tumor cell recognition of γδ T cells.
Role of γδ T in tumor promotion
- The authors should omit this section
Response: We have deleted this section as suggested.
- 176: Does the TME actively transform γδ T cells? Subpopulations exist without influences of TME, so TME probably just favours proliferation/expansion/survival of distinct subsets, but more passively.
Response: We have deleted this sentence as suggested.
- 181: list needs more (primary) references? The authors miss several papers from Silva-Santos, de Visser, Sears labs
Response: We have deleted this sentence as suggested.
- Figure 2: IL-17 not just effect on angiogenesis -> neutrophil recruitment, γδ Treg nomenclature is not well established
Response: We deleted Figure 2 because it is about pro-tumor function of γδ T cells.
- 213: miR-99-5p confusing, is it actually the other one? “miR-99a-5p had no effect on the expression of CD107a, granzyme B and perforin in γδ T cells. Taken together, these results suggest that miR-125b-5p regulates the cytotoxicity of γδ T cells by targeting degranulation and the expression of IFN-γ and TNF-α.” But quite confusing in reference itself as well…so not sure.
Response: We have deleted this sentence as suggested.
- 218: Daley/Miller paper as reference for human patient statement, but paper mainly mouse -> maybe better reference out there? (same for rest of paragraph FoxP3/PD-L1)
Response: We have deleted this sentence as suggested.
Figure 2
- This figure should be omitted
- It is very similar to one published in Nature Reviews Cancer by Silva-Santos
Response: We have deleted this figure because the figure is involved in the pro-tumor function of γδ T cells.
immunotherapy
- 243-245: brief explanation of which specific γδ T cell subpopulation/modification strategy (CAR, transduction with NKR, DOT…) has been exploited for individual trial and outcome of the trial?
Response: We described the related clinical trials of specific γδ T cell subpopulations, particularly Vg9d2 T cells, CAR-γδ T cells and TEG, and summarized the data and advances in Table 2.
- 251: also add a reference?
Response: We deleted this sentence because it is involved in pro-tumor function of γδ T cells.
- 254: “Several studies..” but then just refer to one, following sentence just a repetition?
Response: Thanks. We supplemented references as followed:
Meraviglia S,Eberl M,Vermijlen D et al. In vivo manipulation of Vgamma9Vdelta2 T cells with zoledronate and low-dose interleukin-2 for immunotherapy of advanced breast cancer patients. Clinical and experimental immunology 2010, 161, 290-297.
Wada I, Matsushita H, Noji S, et al. Intraperitoneal injection of in vitro expanded Vγ9Vδ2 T cells together with zoledronate for the treatment of malignant ascites due to gastric cancer. Cancer medicine 2014, 3, 362-375.
- 268: reference 79 is not about Vg9Vd2 – misquoted
Response: Thanks. We deleted this sentence because it is unsuitable in here after modification.
- 273: frequencies of…(2x)
Response: We deleted the example due to unsuited and described relevant content in a new section: “New strategies to improve γδ T cell-based cancer immunotherapy”.
- 277: wording (concerned?)
Response: We changed the description of this sentence as followed. Vδ1 T cells is a subpopulation with strong antitumor activity that is expected to be used in tumor immunotherapy.
- 284: name DOT protocol, established method?
Response: Thanks. We explained DOT definition and introduced established method of DOT as followed. Highly cytotoxic Delta one T (DOT) cells are Vδ1-enriched (>60%) γδ T cells produced by stimulation with TCR agonists and cytokines (IL-4 and IL-15) for over 3 weeks for adoptive cell therapy. A Two-step method was used for DOT cells production: in the expansion stage, sorted γδ T cells were stimulated with IL-4 and TCR agonists for 2 weeks, then followed a differentiation stage with an additional week of IL-15 stimulation. This method can achieve clinical-grade 2000-fold expansion within three weeks, with the percentages of Vδ1 T increase from less than 0.5% to more than 70%. Importantly, NCRs, particularly NKp30 and NKp44, are induced, accompanied with up-regulation of NKG2D and DNAM-1 on expanded DOT cells.
Conclusion
- Repetitive
Response: We deleted repetitive sections and supplemented some new contents.
- Leave out microbiota stuff
Response: We deleted content of microbiota stuff.
Reviewer 3 Report
This is a very comprehensive and well written review.
Author Response
We would like to thank the reviewers and editors for their careful reading and pertinent evaluations on our manuscript. In this version, we revised the manuscript according to the guidance of editor’s and reviewers’ comments. As suggested, we expanded on the anti-tumor properties of the γδ T cells, γδ T-based cancer immunotherapy and multiple new strategies. We deleted the sections of pro-tumor effect of γδ T cells and data related to mouse γδ T cells. We revised the manuscript and corrected the mistakes, including some inaccurate descriptions and some misquoted publications, carefully.
Round 2
Reviewer 2 Report
The authors have greatly improved the manuscript and they have satisfactorily addressed all our comments. The new tables are very good and useful for the community. I still think the authors should use NKR or NCR (but not both) to avoid confusion for the reader.
Author Response
We thank reviewer for the comments, please see the reply below:
The concept for NKR and NCR are different. They can not be replaced each other. NKR means natural killer receptors,
they comprise NKG2D, NCR, DNAM-1, and other receptors. NCR means natural cytotoxicity receptors, include NKp30, NKp44
and NKp46. That is to say, NCRs are members of NKRs. NKRs comprise NCRs and other receptors. Therefore, NKR and NCR
are not the same. They can not be replaced each other.
So, we can not revise the manuscript according to the suggestion from reviewer 2. To avoid the confusion, we revised the sentence in section of Vd1 (3.2) to explain the the correlation between NKR
and NCR in the attached version.